# High sedation needs of critically ill COVID-19 ARDS patients—A monocentric observational study

Armin Niklas Flinspach[1]*, Hendrik Booke[1], Kai Zacharowski[1], Ümniye Balaban[2], Eva Herrmann[2], Elisabeth Hannah Adam[1]

1 Department of Anaesthesiology, Intensive Care Medicine and Pain Therapy, University Hospital Frankfurt, Goethe-University Frankfurt, Hessen, Germany, 2 Department of Biostatistics and Mathematical Modelling, Goethe-University Frankfurt, Hessen, Germany

* armin.flinspach@kgu.de

**Data Availability Statement:** Data cannot be shared publicly. The datasets generated and/or analyzed during the current study are not publicly available due to national data protection laws but

## Abstract

### Background

Therapy of severely affected coronavirus patient, requiring intubation and sedation is still challenging. Recently, difficulties in sedating these patients have been discussed. This study aims to describe sedation practices in patients with 2019 coronavirus disease (COVID-19)-induced acute respiratory distress syndrome (ARDS).

### Methods

We performed a retrospective monocentric analysis of sedation regimens in critically ill intubated patients with respiratory failure who required sedation in our mixed 32-bed university intensive care unit. All mechanically ventilated adults with COVID-19-induced ARDS requiring continuously infused sedative therapy admitted between April 4, 2020, and June 30, 2020 were included. We recorded demographic data, sedative dosages, prone positioning, sedation levels and duration. Descriptive data analysis was performed; for additional analysis, a logistic regression with mixed effect was used.

### Results

In total, 56 patients (mean age 67 (±14) years) were included. The mean observed sedation period was 224 (±139) hours. To achieve the prescribed sedation level, we observed the need for two or three sedatives in 48.7% and 12.8% of the cases, respectively. In cases with a triple sedation regimen, the combination of clonidine, esketamine and midazolam was observed in most cases (75.7%). Analgesia was achieved using sufentanil in 98.6% of the cases. The analysis showed that the majority of COVID-19 patients required an unusually high sedation dose compared to those available in the literature.

### Conclusion

The global pandemic continues to affect patients severely requiring ventilation and sedation, but optimal sedation strategies are still lacking. The findings of our observation suggest

are available upon reasonable request from the corresponding author, or via the data protection officer of the University Hospital Frankfurt (Datenschutz@kgu.de).

**Funding:** The author(s) received no specific funding for this work.

**Competing interests:** KZ has received honoraria for participation in advisory board meetings for Haemonetics and Vifor and received speaker fees from CSL Behring and GE Healthcare. KZ is the Principal Investigator of the EU-Horizon 2020 project ENVISION (Intelligent plug-and-play digital tool for real-time surveillance of COVID-19 patients and smart decision-making in Intensive Care Units). None of the Competing Interests mentioned is related to the present work. The author confirms that the disclosed conflicts of interest of KZ does not alter the adherence to PLOS ONE policies on sharing data and materials. There are no patents, products in development or marketed products associated with this research to declare.

unusual high dosages of sedatives in mechanically ventilated patients with COVID-19. Pre-scribed sedation levels appear to be achievable only with several combinations of sedatives in most critically ill patients suffering from COVID-19-induced ARDS and a potential association to the often required sophisticated critical care including prone positioning and ECMO treatment seems conceivable.

## Introduction

Approximately 5% of COVID-19 infections are associated with COVID-19-induced acute respiratory distress syndrome (C-ARDS). The pandemic poses a major challenge to health care systems because of the need for intensive care therapy and mechanical ventilation including sedation. The sedation required for elaborate critical care treatment in patients with C-ARDS, including prone positioning and veno-venous extracorporeal membrane oxygenation (vvECMO) therapy, has already been discussed as a sophisticated task [1]. Thus far, limited data for sedation in patients suffering from C-ARDS are available. Recently, Wongtangman et al. published a first retrospective comparison between ARDS patients with and without causative COVID-19 pneumonia. They demonstrated a significantly increased need for sedation and analgesics on the basis of a sedative burden index [2]. In addition, Kapp et al. were able to demonstrate a significant association between sedation depth and mortality [3].

It is unclear whether the numerous recommendations on sedation concepts published for patients with acute respiratory distress syndrome (ARDS) are appropriate for patients with C-ARDS [4–7]. Avoidance of deep sedation during intensive care is clearly recommended for patients with Non COVID-19 ARDS whenever possible. An exception to these recommendations is the occasional need for deep sedation when performing advanced therapies in severe ARDS, such as improved patient-ventilator synchrony, prone positioning and vvECMO [8, 9]. However, daily interruptions of continuous sedation are highly recommended, which in turn shortens the duration of mechanical ventilation and subsequently the length of stay in the ICU, leading to fewer complications [9, 10]. In the absence of a coronavirus-specific therapy, the recommendations are focused on protective lung ventilation and positioning therapy. However, this often appears to be unattainable by a single combination of a hypnotic and an opioid, so a sedative strategy with multiple drugs is required. To quantify this issue, we assessed the analgesia and sedation of all critically ill COVID-19 patients admitted to our institute.

## Material and methods

This is a retrospective observational study at the University Hospital Frankfurt, which has been approved by the institutional ethics board of the University of Frankfurt (#20–643). The need for informed consent from individual patients was waived due to the context of the study being a sole retrospective review. This manuscript adheres to the applicable CONSORT guidelines.

All patients were treated according to the recommended ABCDE therapy bundle [11]. The individual pharmaceutical therapy was determined by the attending physician.

### Patient population

We included all patients admitted to the intensive care unit between April 4, 2020, and June 30, 2020 who were already diagnosed with severe acute respiratory syndrome coronavirus type

2 (SARS-CoV-2) infection or tested positive during treatment [12]. The patients' medical records were assessed between June 2020 and July 2020 and completed through database access by our research group by August 2020.

Medication was primarily administered in analogy to the in-house standard of our 32-bed ICU for critically ill non-COVID-ARDS patients. Accordingly, continuous intravenous (iv.) application of a strong opioid (e.g., sufentanil) in combination with continuous iv. application of a sedative has been applied. Primary sedatives were propofol or clonidine and in case of sedation difficulties a combination with midazolam was used. In the case of primary use of propofol, conversion to clonidine was initiated in the case of a prolonged therapy to avoid propofol infusion syndrome.

The application of adjunct agents such as barbiturates or antipsychotics is not practiced in our department. In analogy to the existing standards of our ARDS center and the ABCDE guidelines, if necessary and sufficient analgesia is given, an escalation with further sedatives is performed at the decision of the attending physician [11]. Neuromuscular blocking agents were not used as standard but in case of uncontrollable ventilator asynchrony. All patients received mechanical ventilation using an Elisa 800 (Löwenstein Medical, Bad Ems, Germany) or Hamilton G5 (Hamilton Medical, Bonaduz, Switzerland) ICU ventilator, as well as intensive care therapy, according to the current recommendations for the treatment of C-ARDS [8, 13, 14]. Exclusion criteria were the absence of invasive ventilation and consecutive sedation or a duration of ventilation less than 24 hours. The use of volatile sedation also led to study exclusion.

The observation period began with intubation and corresponding sedation or with the admission of patients that were already intubated. The observation period ended with death, tracheostomy, or cessation of pharmaceutical sedation after a successful spontaneous breathing trial and subsequent extubation. In order to exclude short-term deepening of sedation, e.g. bolus application for interventional procedures, we only considered continuously sedation regimes of more than four hours of continuous application for analysis. In accordance with our standards to determine adequate sedation a bedsige examination was carried out by the attending physician, in addition to an evaluation of the reliable Richmond Agitation and Sedation Scale (RASS), which assessed ventilator synchrony, signs of stress and the occurrence of vegetative agitation [15]. Adequate ventilator synchrony was defined as the clinically predominant absence of asynchronous phases, which was based on the observation of respiratory volume pressure curves by the attending staff [16]. Following published recommendations, a target RASS of 0 to -1 was aimed for in the therapy standard [4]. In prone position and for ECMO therapy a RASS of -3 to -4 was targeted for sufficient psycho-vegetative protection [10, 17].

## Data collection

Clinical data were continuously recorded using a patient data management system (PDMS; Metavision 5.4, iMDsoft, Tel Aviv, Israel). We recorded demographic data, sedative and analgetic dosages, clinical satisfaction of sedation levels, RASS, positioning therapy, vvECMO therapy and outcomes (death or discharge).

## Statistical analysis

No statistical power calculation was conducted prior to this retrospective study. The present study is a retrospective analysis. Data with continuous scale are represented as mean (± standard deviation), data with categorical scale are presented as frequencies and percentages. Additionally, spontaneous breathing time and RASS values were analyzed using logistic regression

mixed effect models using a correlation matrix with autoregressive and moving average process for longitudinal binary data.

All statistical tests were two-tailed and results with p ≤0.05 were considered statistically significant. All calculations/analyses were performed with SPSS (IBM Corp., Version 26, Chicago, IL, USA) or R for Statistical Computing (The R Foundation, Version 4.0, Vienna, Austria). The packages 'MASS' and 'nlme' were used [18, 19].

## Results

During the study period we assessed 85 patients and were able to acquire a data set for the evaluation for 56 of them (**Fig 1**). The demographic and clinical characteristics of patients at the time of admission are presented in **Table 1**.

### CONSORT - Sedation of critically ill COVID-19 patients

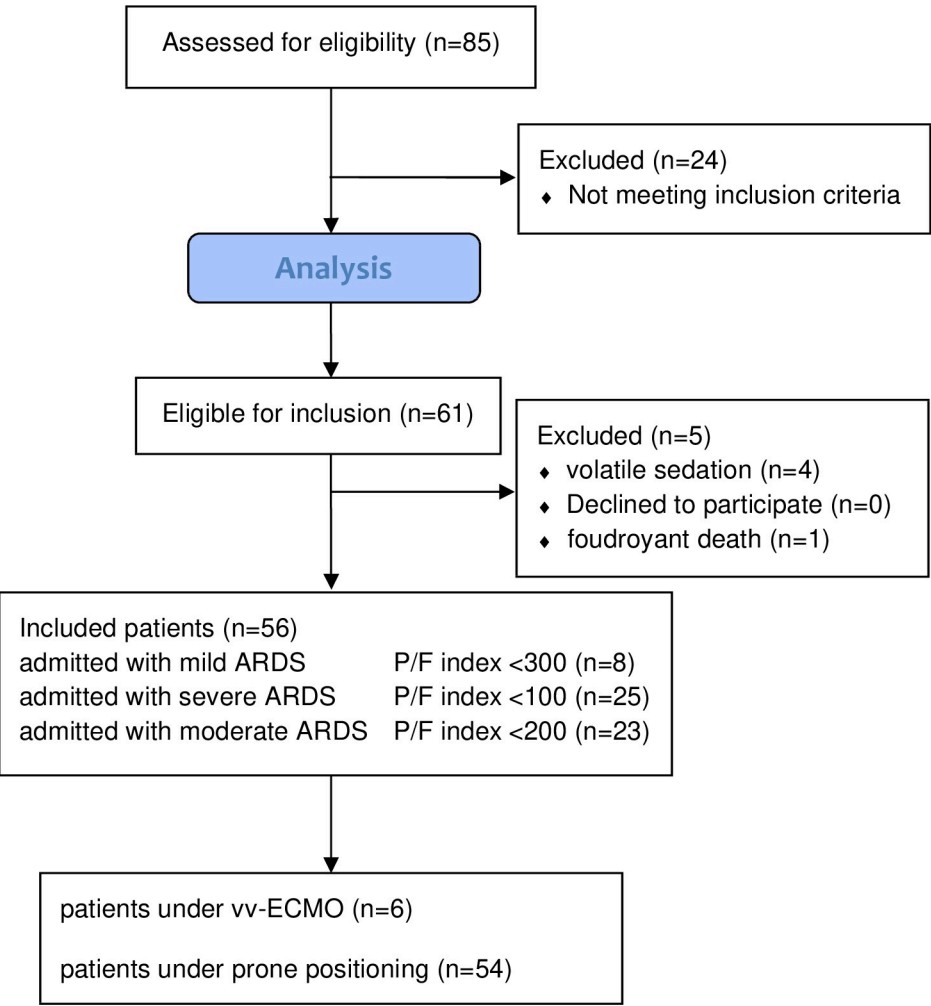

**Fig 1. Flow chart of patients included into the study (according to the CONSORT criteria).** ARDS = acute respiratory distress syndrome, P/F index = Horovitz Oxygenation index = $p_aO_2/F_iO_2$, vv-ECMO = veno-venous-extra corporeal membrane oxygenation.

**Table 1. Clinical characteristics of C-ARDS patients.**

| patients included | n = 56 | |
|---|---|---|
| **Characteristics** | | |
| Age, y | 67 | (14.0) |
| Sex, m | 43 | [76.8%] |
| bodyweight, kg | 95.93 | (21.53) |
| BMI | 31.66 | (6.71) |
| SAPS II | 43.88 | (10.84) |
| Oxygenation index at admission | 145.21 | (66.16) |
| Hospital stay before ITN, d | 3.33 | (5.08) |
| Median observation period, h | 224 | (139.5) |
| Prone positioning | n = 54 | [96,4%] |
| Median treatment time, h | 95.0 | (32.3) |
| vvECMO treatment | n = 6 | [10.7%] |
| Median treatment time, h | 252 | (191.3) |
| cRRT treatment due to AKI | n = 27 | [48.2%] |
| Median treatment time, h | 152 | (125) |
| Mortality | n = 26 | [46.4%] |

Table 1: Data are presented as mean (SD) or as patient number [percentage] where applicable.

Abbreviations: AKI = acute kidney injury, BMI = Body mass index, cRRT = continuous renal replacement therapy, d = days, h = hours, ITN = intubation kg = kilogram, Oxygenation index = paO2/FiO2, SAPS II = Simplified Acute Physiology Score II, SD = Standard deviation, vvECMO = veno-venous extracorporeal membrane oxygenation, y = years

The continuous analgesia was performed with sufentanil (0.13 ($\pm$0.09) $\mu g \cdot kg^{-1} \cdot h^{-1}$) throughout 98.6% of the observation period (in 1.4% with remifentanil (0.15 ($\pm$0.06)) $\mu g \cdot kg^{-1} \cdot min^{-1}$). Using a single sedative agent in combination with a strong opioid achieved prescribed sedation level in 38.1% of the cases. Of all patients, five were satisfactorily sedated with a single hypnotic during the entire treatment period. We found that 48.8% of patients required a double sedation regimen to achieve satisfactory sedation. The majority (59.3%) of these patients received a combination of clonidine and midazolam. Triple sedation combined with an opioid was used to achieve satisfactory sedation in 12.8% (1742 h) of patients being treated for C-ARDS. In 75.7% of these patients, triple sedation was administered with a combination of midazolam, clonidine and esketamine. In one patient, a temporary (48 h) four-fold sedation was required. The hypnotics used for single or multiple sedation are presented in **Figs 2** and **3**. The use of the short-acting substances dexmedetomidine and lormetazepam were recorded in 7.4% and 15.3% total treatment time, respectively. The documented overall dosages for the central $\alpha_2$ inhibitors clonidine were 1.54 ($\pm$0.79) $\mu g \cdot kg^{-1} \cdot h^{-1}$ and dexmedetomidine 0.54 ($\pm$0.58) $\mu g \cdot kg^{-1} \cdot h^{-1}$, respectively. For the gamma-aminobutyric acid (GABA) receptor active benzodiazepines we recorded, midazolam 0.86($\pm$0.76) $mg \cdot kg^{-1} \cdot h^{-1}$ and lormetazepam 0.013 ($\pm$0.023) $\mu g \cdot kg^{-1} \cdot min^{-1}$, each as a mean dosage, and for propofol 1.66($\pm$1.40) $mg \cdot kg^{-1} \cdot h^{-1}$ was administered. For esketamine as N-methyl-D-aspartate (NMDA) receptor inhibitor a mean dose of 0.86($\pm$0.76) $mg \cdot kg^{-1} \cdot h^{-1}$ was found. A complete list of dosages and combined use can be found in **S1 Table**.

Sedation depth was assessed using the RASS. The graduated depth of sedation over the treatment period is shown in **Fig 4**.

All patients worsened their oxygenation index below <200 during treatment, representing moderate or severe ARDS, leading to prone positioning assuming positive effects.

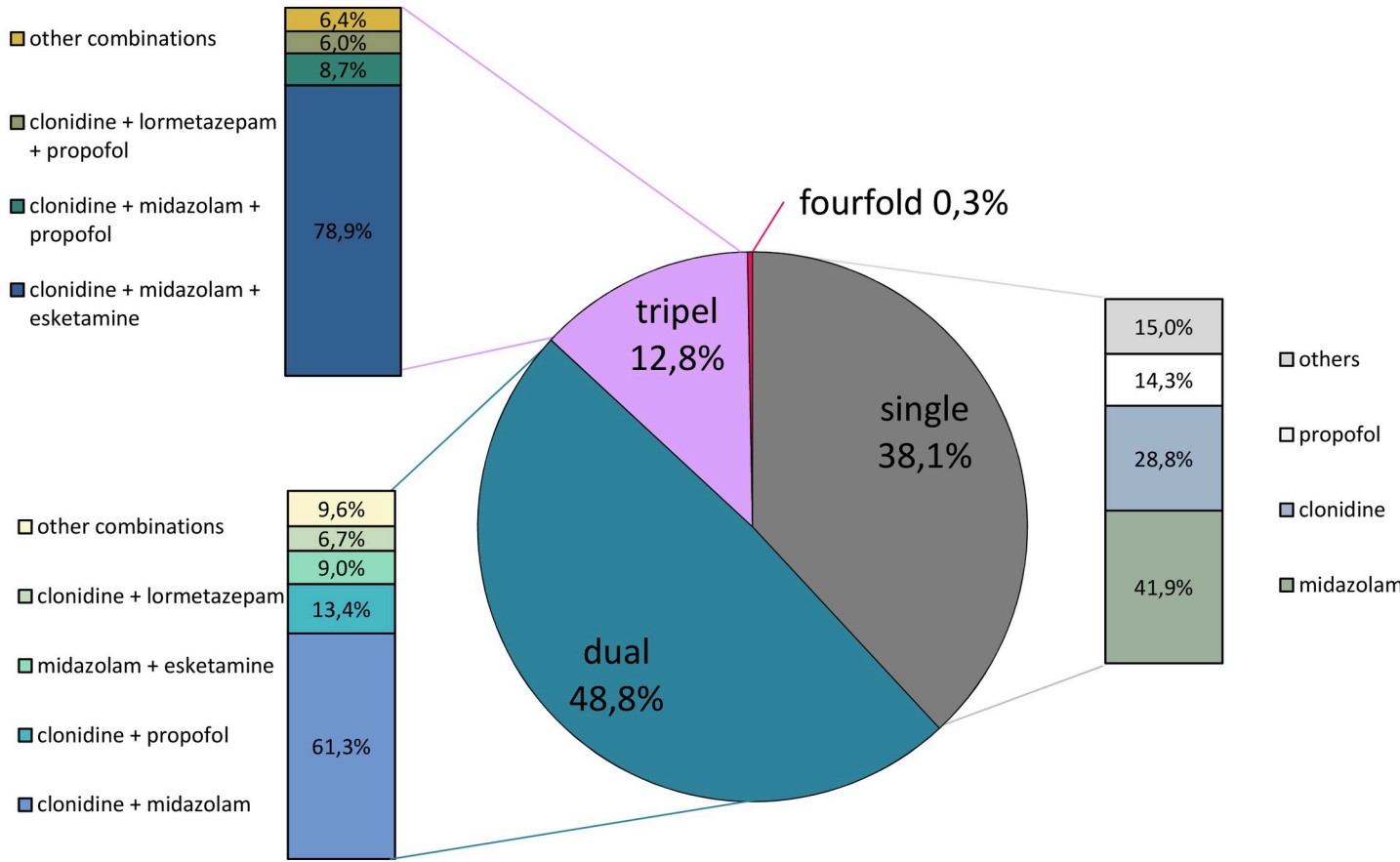

**Fig 2. Single and multiple sedation in COVID-19 patients.** The pie chart represents the types and frequency of single or multiple sedation in percentage. The associated bar charts represent the subdivision of the applied sedatives and their combinations in percentage. *other combinations: sum of conceivable otherwise twofold or threefold combined sedative applications.

A logistic regression analysis with mixed effect of the measured sedation revealed a significant association of low RASS (RASS ≤ -3, p < 0.05) with prone vs. supine positioning therapy. Furthermore, we were able to show that patients undergoing vv-ECMO therapy needed deep sedation (RASS ≤ -4) more often (p < 0.05) than patients without. Logistic regression revealed a significant decrease in spontaneous ventilation during prone positioning compared to supine position (832 vs. 1384 h; p = 0.05).

In total, we observed 26 deaths, of which 20 patients dropped out of the study due to death. We could not find any correlation between the observed sedation depth or required sedation amount and patient survival (data not shown).

In our study cohort, 35 patients received a single dose (5 cisatracurium and 30 rocuronium) of neuromuscular blocking agents (NMBAs). In 15 of the cases NMBA were applied for endotracheal intubation in the remaining to treat uncontrollable coughing and to improve adequate ventilator synchrony. In addition, three patients required continuous administration of cisatracurium for a total of 592 hours (mean: 120 [109]). Among these, two received vv-ECMO treatment. In nine patients dilatatory tracheostomy was performed. Tracheostomy was conducted after a mean of 20 ±6.46 days of ventilatory support. For two patients, a prone positioning was not possible due to super obesity (body mass index ≥50).

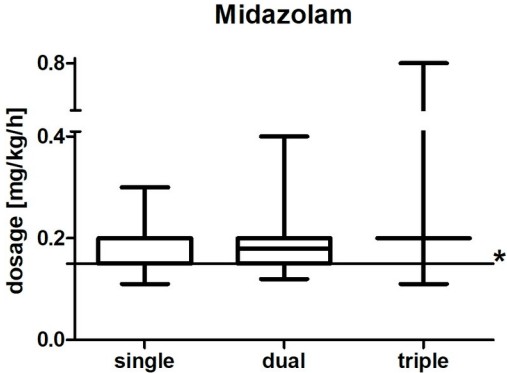

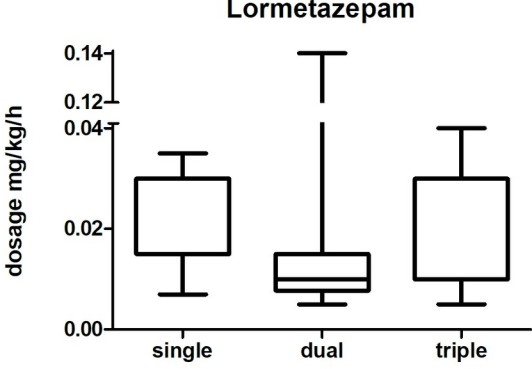

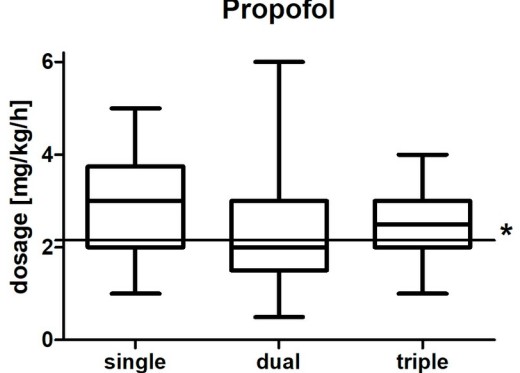

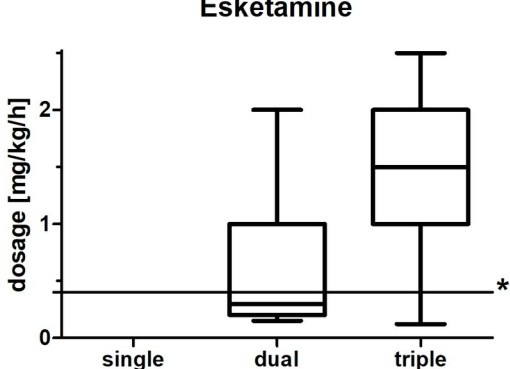

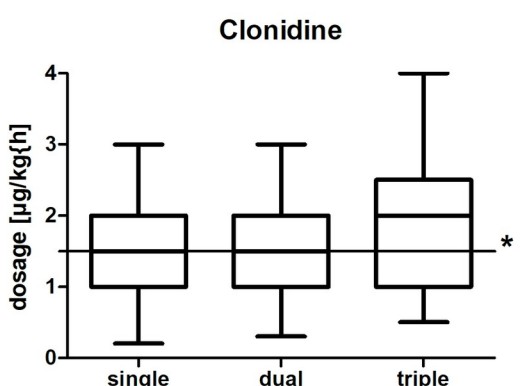

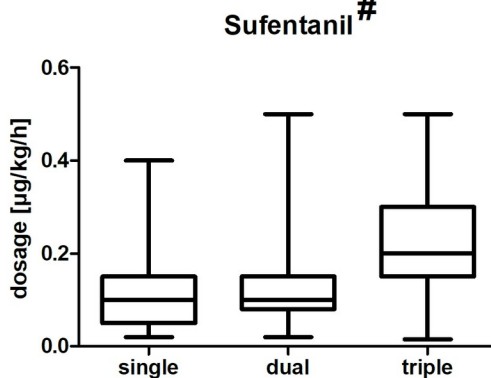

**Fig 3. Pharmaceutical dosages administered.** Administered pharmaceutical dosages in the time interval of the applied single, double or triple substance use. Data presented as box-whisker plots. # = Applied dosage of sufentanil as an opioid for analgesic therapy under prescribed sedation level. * = literature based median dosage. single = use of the corresponding substance as monosedativ, dual = application in combination with another sedative, triple = use in combination with two further sedatives, µg = microgram, mg = milligram, kg = kilogram, h = hour, min = minute.

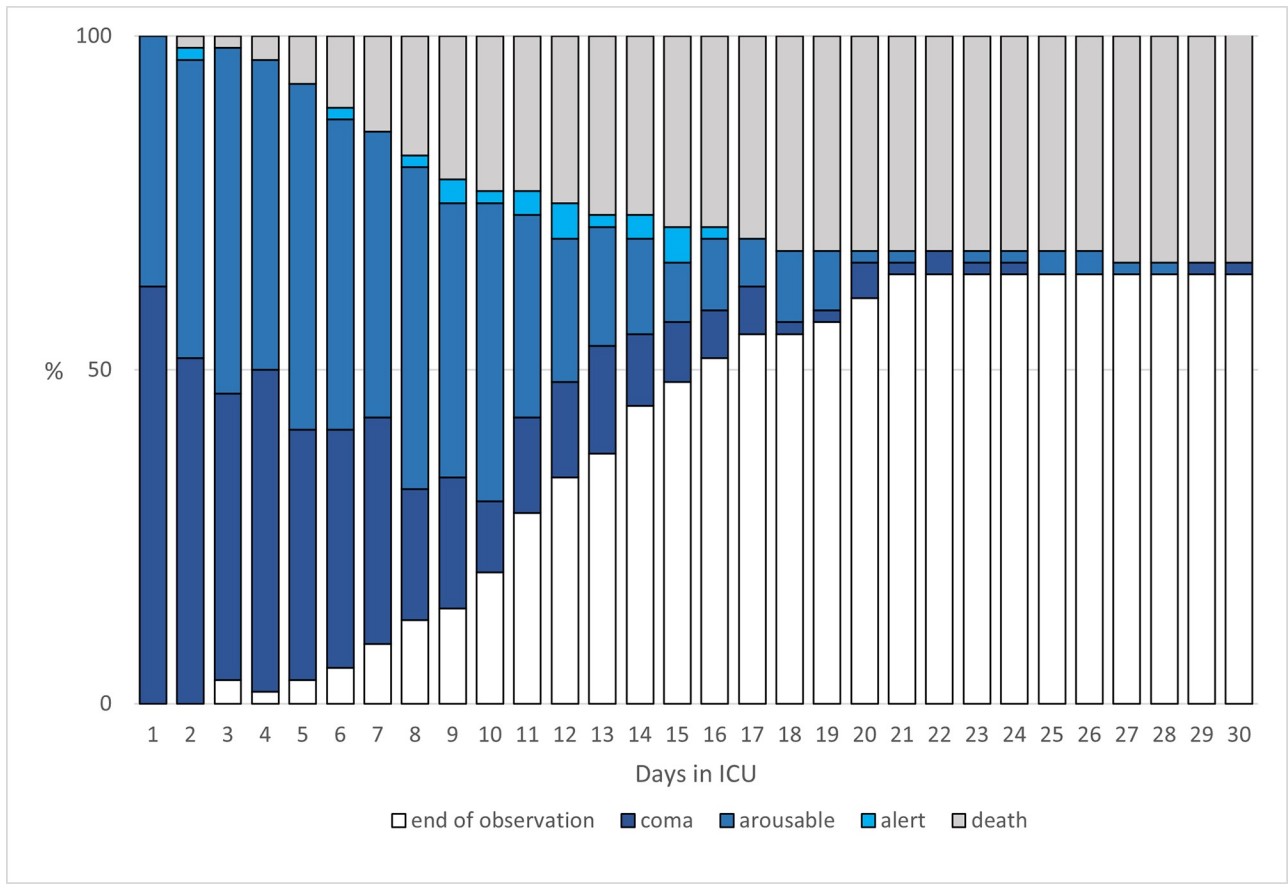

**Fig 4. Cumulative frequency of observed sedation depth.** Graphical plot of the cumulative frequency of observed sedation depth and death in patients with moderate or severe COVID-19-related acute respiratory distress syndrome (ARDS) requiring mechanical ventilation. Sedation depth measured by Richmond Agitation Sedation Score (RASS) is graphically represented by coma (dark blue, RASS ≤ -3), arousable (blue, RASS = -2), or alert (light blue, RASS ≥ -1). Furthermore, leaving the observation period by death (in gray) or tracheostomy or extubation (white) is graphically represented.

We did not observe any protracted delirium in the patients we observed during the study period. Also, no disproportionate rate of nosocomial infections or cardiovascular complications leading to prolonged sedation were observed.

## Discussion

In our retrospective observational study, we found evidence of unusually high sedative medication requirements as well as of multiple use as combination therapies, leading to a challenging sedation in patients with moderate to severe C-ARDS. Our results are consistent with the repeatedly raised suspicion of aggravated sedation with previously published results from other study groups [2, 3]. In our study, we found a high mortality (46.6%) comparable to international data among mechanically ventilated severely affected C-ARDS patients [20].

For mono-sedation, patients were mainly treated with the benzodiazepine midazolam or the central $\alpha_2$-agonist clonidine. When combination therapy was required, these two substances were usually used, and in most cases of triple sedation, esketamine was added. To date, it has been shown worldwide that patients with C-ARDS often need ventilation for a longer period of time [1, 21, 22]. Although, in our study, the attending physician was free to decide which sedative substance to administer, long-acting rather than short-acting sedatives were chosen, presumably due to the expected prolonged ventilation time.

Clonidine at a mean dosage of 1.5 μg·kg$^{-1}$·h$^{-1}$ has been shown to work as sufficient mono-sedation in various studies [23–26]. With regard to the use of central α$_2$-agonists, such as clonidine and dexmedetomidine, it should be emphasized that these intrinsically potent sedatives are beneficial in combination therapies due to their well-known co-analgesic and co-sedating properties [27, 28]. Thus, the frequent use of central α$_2$-agonists in high dose for mono (28,8), double (59.3%) and triple (75.7%) sedation during this observation is well-founded, but also illustrates the complexity of achieving prescribed sedation level in these patients.

The required dosages of esketamine (mean 0.86 (±0.75) mg·kg$^{-1}$·h$^{-1}$) observed in our study were unusually high compared to the dosages referred to in the literature (mean approximately 0.4 mg·kg$^{-1}$·h$^{-1}$) [29, 30]. This observation is especially meaningful when taking into account that esketamine was only administered in combination with a sedative that had already been used at its optimum dose. Such frequent use of esketamine as a substance rarely used in modern intensive care was similarly found in another study [2]. For effective sedation with midazolam, a mean dosage of 0.15 (±0.1) mg·kg$^{-1}$·h$^{-1}$ was found necessary in several studies [31–34]. In our patient cohort, midazolam was used at a similar mean dosage 0.14 (±0.10) mg·kg$^{-1}$·h$^{-1}$ to obtain prescribed sedation. Due to the risk of a propofol infusion syndrome during long-term ventilation, propofol was only administered for a maximum of a few days in our patients. Nevertheless, we observed an increased propofol mean dosage of 2.50 (±0.96) mg·kg$^{-1}$·min$^{-1}$, in comparison to the mean dosages reported in the literature of 2.15 mg·kg$^{-1}$·min$^{-1}$ [33, 35, 36].

Few data exist to date to compare in-hospital sedation management strategies in the setting of COVID-19 patient care. However, our in-hospital standards appear to be consistent with previously published regimens for severe COVID-19 patients in terms of the classes of agents used. Previously published reports also describe the predominant use of propofol, benzodiazepines, central α$_2$-agonists and potent opioids, as well as the use of esketamine to achieve the prescribed depth of sedation [2, 3].

In this study sufentanil was evaluated as a primary analgesic rather than a sedative agent. However, it should be noted that sufentanil has significant sedative properties, which highlights the complexity of sedation and stresses the high sedation requirement given the 98.6% use during the study period. Due to the predominantly long positioning periods, analgesia monitoring could not be performed continuously using a validated score. Sufentanil was administered primarily to facilitate endotracheal tube tolerance and for positioning maneuvers. We were unable to detect a significant increase in the required sedatives for prone positioning.

The observations frequently expressed so far, confirm that patients with C-ARDS pose major challenges in regard to the feasibility of sedation, especially to enable prone positioning or vv-ECMO treatment [1, 3]. Younger age has been speculated to be a key factor and reasoning as to the higher sedation doses required, which may impede the achievement of prescribed sedation level. Our observations do not support this hypothesis, as the average age of our patients was 67 years and we still encountered difficulties in regard to our sedation regimen [1]. Besides sedatives and opioids, neuromuscular blockade is recommended for ARDS therapy in the first 48 hours after intubation [37–39]. The notably rare application of NMBAs might be explained by the fact that 22 patients were secondarily transferred to our COVID-19 center, meaning the initial treatment phase took place beforehand. The majority of NMBA applications were conducted to resolve therapy-resistant ventilator asynchronies in spontaneous breathing mode.

In contrast to the unusually high sedation doses needed, we still observed a high rate of ventilator-assisted spontaneous breathing, aiming to improve oxygenation and reduce diaphragmatic muscle loss in line with the literature [40, 41]. However, spontaneous breathing in

patients with ARDS should be on a case-by-case decision. In the future, considering the challenging sedation requirements of patients with COVID-19, the use of volatile anesthetics should be considered in appropriate cases, in addition to the early use of combined sedatives [42].

Some limitations must be taken into account when interpreting our results. A potential limitation to the internal validity arises from a sampling bias in the study population: In our study we included a substantial number of patients referred to our hospital by primary care providers. This may have led to an assessment of patients who were more severely affected by COVID-19 compared to the general COVID-19 population. We also included ECMO patients, although the use of sedatives in ECMO has been described to be higher during the beginning of a ECMO run [43]. We could not determine to what extent this affects an increased dosage in prolonged ECMO applications under C-ARDS. Furthermore, we investigated a wide range of different sedatives in multiple combinations, which limits the generalizability of the observations. Moreover, we did not investigate whether a history of drugs or alcohol was present, which might have influenced the administration of any sedatives. The interpretation of our data is based on published reference values and must therefore be considered with the limitation of a missing non-C-ARDS control group. However, our findings are consistent with the frequently expressed observation of a massively increased need for sedation in critically ill COVID-19 patients. The authors feel confident, that the observations obtained within this study are applicable to patients suffering from COVID-19 requiring critical care therapy.

However, it remains unclear what causes the impaired sedation. Nevertheless, early hyposmia was described as a characteristic symptom of COVID-19 [44]. In the meantime, it could be demonstrated that the novel corona virus by far does not only causes an infection of the lungs, but can also affect the central nervous system in addition many other organs. Especially in severely affected COVID-19 patients with viremia, an alteration of the CNS is conceivable [45]. Thus, the aggravated sedation could occur as a consequence of an infection of the central nervous system.

Future studies should address the underlying reasons for the observed high sedative medications required in patients with C-ARDS.

## Conclusion

The global pandemic continues to affect patients severely, leading to the necessity of ventilation and sedation, but optimal sedation strategies are still lacking. The findings of our observation suggest unusual high dosages of sedatives in mechanically ventilated patients with COVID-19. Prescribed sedation levels appear to be achievable only with multiple combinations of sedatives in most critically ill patients suffering from C-ARDS and a potential association to the often required sophisticated critical care including prone positioning and ECMO treatment seems conceivable.

## Supporting information

**S1 Table. Applied sedatives and analgetic dosages.**
(DOCX)

## Author Contributions

**Conceptualization:** Armin Niklas Flinspach, Hendrik Booke, Elisabeth Hannah Adam.

**Data curation:** Armin Niklas Flinspach, Hendrik Booke.

**Formal analysis:** Armin Niklas Flinspach, Ümniye Balaban, Eva Herrmann.

**Investigation:** Elisabeth Hannah Adam.

**Methodology:** Armin Niklas Flinspach.

**Project administration:** Armin Niklas Flinspach.

**Software:** Ümniye Balaban, Eva Herrmann.

**Supervision:** Kai Zacharowski, Eva Herrmann.

**Visualization:** Armin Niklas Flinspach.

**Writing – original draft:** Armin Niklas Flinspach, Elisabeth Hannah Adam.

**Writing – review & editing:** Armin Niklas Flinspach, Kai Zacharowski, Elisabeth Hannah Adam.

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
