## [Decision Letter · Decision Letter 0]

12 May 2021

PONE-D-21-08369

High sedation needs of critically ill COVID-19 ARDS patients- a monocentric observational study.

PLOS ONE

Dear Dr. Frankfurt,

Thank you for submitting your manuscript to PLOS ONE. After careful consideration, we feel that it has merit but does not fully meet PLOS ONE’s publication criteria as it currently stands. Therefore, we invite you to submit a revised version of the manuscript that addresses the points raised during the review process.

We look forward to receiving your revised manuscript.

Kind regards,

Chiara Lazzeri

Academic Editor

PLOS ONE

Journal Requirements:

2. Thank you for providing the date(s) when patient medical information was initially recorded. Please also include the date(s) on which your research team accessed the databases/records to obtain the retrospective data used in your study.

'EHA received a research grant of the German Research Foundation (AD 592/1-1)

KZ received financial support from a multitude of companies, a detailed list is attached to the manuscript.

The further authors declare that there are no conflicts of interest. '

a. Please confirm that this does not alter your adherence to all PLOS ONE policies on sharing data and materials, by including the following statement: "This does not alter our adherence to  PLOS ONE policies on sharing data and materials.” (as detailed online in our guide for authors http://journals.plos.org/plosone/s/competing-interests).  If there are restrictions on sharing of data and/or materials, please state these.

Please note that we cannot proceed with consideration of your article until this information has been declared.

5. Please include captions for your Supporting Information files at the end of your manuscript, and update any in-text citations to match accordingly. Please see our Supporting Information guidelines for more information: http://journals.plos.org/plosone/s/supporting-information

Reviewers' comments:

Reviewer's Responses to Questions

**Comments to the Author**

1. Is the manuscript technically sound, and do the data support the conclusions?

Reviewer #1: Partly

Reviewer #2: Yes

2. Has the statistical analysis been performed appropriately and rigorously? 

Reviewer #1: Yes

Reviewer #2: Yes

3. Have the authors made all data underlying the findings in their manuscript fully available?

Reviewer #1: Yes

Reviewer #2: Yes

4. Is the manuscript presented in an intelligible fashion and written in standard English?

Reviewer #1: Yes

Reviewer #2: Yes

5. Review Comments to the Author

Reviewer #1: The goal of the authors was to describe sedation practices in patients with COVID ARDS in their medical center. They propose that their patients received unusually high doses of sedatives. Given that this is a descriptive study, I propose that some descriptions need to be extended/more granular, for a practicing clinician to derive value from this study.

Concerns

1. The authors did not compare the COVID-19+ ARDS cohort to cohort with nonCOVID- ARDS to support their claim that COVID+ patients require higher doses of sedatives. Such comparisons are feasible and were already published by other authors (e.g., Wongtangman K, Santer P, Wachtendorf LJ, Azimaraghi O, Baedorf Kassis E, Teja B, Murugappan KR, Siddiqui S, Eikermann M; SICU Optimal Mobilization Team (SOMT) Group. Association of Sedation, Coma, and In-Hospital Mortality in Mechanically Ventilated Patients With Coronavirus Disease 2019-Related Acute Respiratory Distress Syndrome: A Retrospective Cohort Study. Crit Care Med. 2021 Apr 5. doi: 10.1097/CCM.0000000000005053. Epub ahead of print. PMID: 33861551.) The authors need to reference this published study in the Introduction and discuss how their observations are unique from this published study.

2. Authors state they they recorded “sedation levels” but these are not presented in the figures or tables. A Figure which would display daily sedation levels with daily doses of sedatives throughout the duration of mechanical ventilation would strengthen the paper. I assume that sedation levels decrease (i.e. RASS moves towards positive values) as survivors approach extubation/tracheostomy but RASS may/may not decrease in nonsurvivors. How do daily drug doses differ in survivors/nonsurvivors as they progress through the ICU stay? Given extremely high mortality in this cohort (48%) such comparison could be feasible. An interesting finding would be relatively low exposure to sedatives, but deep coma, in nonsurvivors – suggesting other factors (e.g. metabolic, inflammatory) in driving the coma.

3. Authors state that their ARDS patients are primarily transferred patients that were initially managed in other institutions. Were the drug exposures in outside institutions analyzed? At what day of mechanical ventilation were these patients transferred? As patients are “stabilized” in the receiving institution, changes to sedatives are oftentimes made. Additionally, oversedation in outside institution will lead to tolerance and the receiving institution “inherits” opioid-tolerant and hypnotic-tolerant patients. I suppose the authors were able to make many positive adjustments to sedation towards weaning sedation in the first 48 hours after they received the patients. These data would be novel and worth to report.

4. Authors report that 96% patients were ventilated in prone position. Authors also state that they targeted deeper levels of sedation (RASS-4) in prone position. This means that even some patients with mild and probably all with moderate ARDS were proned in this center. This may seem aggressive proning approach to some other centers. Did patients with mild and moderate ARDS progress to severe, and that’s why they were proned? This needs to be discussed. Otherwise, the whole study could be viewed as a cohort of aggressive proning and associated/justifiable deep sedation. There needs to be more granularity in reporting the sedation doses – prone vs. supine position, daily doses, survivors vs nonsurvivors, etc.

5. Authors use variable terminology “ feasible sedation level”, “adequate sedation level” , “appropriate sedation”, “sufficient analgesia” . The meaning of these is unclear. I propose to use “prescribed sedation level” and “ actual sedation level” . For instance, physician prescribes the level of RASS 0, but on exam, finds that patients has level of RASS-4. Analgesia did not seem to be evaluated in this study (e.g. with CPOT scale), therefore it is unclear why authors comment on sufficient analgesia. In general, COVID-19 is not a pain-producing condition unless it is associated with thromboses (MI, mesenteric ischemia, limb ischemia,…).

6. Who were the patients that required highest levels of ketamine and sufentanil or the combination of 3 sedatives in this cohort? Were all these ECMO patients? Did all these patients die? If patients with highest sedation doses actually survived, this could teach us that very high sedation requirements may not necessarily be followed by bad outcomes. This would be informative.

7. Authors state that they try to avoid propofol. “Due to the risk of a propofol infusion syndrome during longterm ventilation, propofol was only administered for a maximum of a few days in our patients”. Then they state in Methods “propofol or clonidine and in case of sedation difficulties a combination with midazolam is used”. This manuscript needs to be more consistent, if reader is supposed to understand the sedation practices in author institution.

Reviewer #2: Dr. Frankfurt, et al have submitted a retrospective chart review of COVID-associated ARDS patients and the sedation management strategies used for these patients. The manuscript highlights how these patients may require higher than average doses of IV sedatives compared to other ICU patients. The structure of the manuscript is appropriate. The logic is clear and relatively well-characterized. The figures and data analyses are appropriately displayed.

COMMENTS:

1. Please review your manuscript more thoroughly for grammar, punctuation, and syntax errors. There were numerous examples of this throughout the text, especially in the figure descriptions.

2. In the discussion, the analysis of your results is stated well; however I believe some further discussion about how your results compare to others is warranted. Additionally, any hypothesis or speculation regarding why higher average doses are required should be elaborated upon further. Is it due to the nature of ARDS or are other pathophysiologic factors to be considered as well?

3. The most common combination of your hospital's IV sedative regimen should be compared to others published in the literature if possible. Otherwise, the generalizability of your results is in question as higher doses may have been required due to non-ideal practices. For instance, other hospitals use dexmedetomidine much more readily than clonidine infusions due to its purported benefits with alpha-2 receptor selectivity and delirium prevention.

4. An analysis of other factors that may have caused higher than average mean doses should be considered as well. Did patients have protracted delirium? Did other complications occur that led to more days of sedation?

Thank you for your submission and your hard work!

6. PLOS authors have the option to publish the peer review history of their article (what does this mean?). If published, this will include your full peer review and any attached files.

Reviewer #1: No

Reviewer #2: No

---

## [Author Response · Author response to Decision Letter 0]

8 Jun 2021

Dear Editor,

The authors sincerely thank the editorial board of PLOSOne for their time and expert guidance in reviewing our manuscript. We have carefully considered their concerns about necessary changes and have modified the manuscript accordingly.

In addition, I would like to point out that I made a mistake during the registration in the manuscript editor. There was a slip in the line, so that my last name was replaced by the location of the department. In the meantime, this issue was corrected in the PLOSOne manuscript editor registration by myself.

Enclosed we would like to submit to you the detailed/point by point responses to the PLOS ONE editors' comments.

Sincerely,

Armin Flinspach, MD, DESA

Editorial Comments:

The manuscript was modified according to the PLOSOne formatting and styling requirement.

 2. Thank you for providing the date(s) when patient medical information was initially recorded. Please also include the date(s) on which your research team accessed the databases/records to obtain the retrospective data used in your study.

The manuscript was updated to include the additional information that the patients' medical records were evaluated between June 2020 and July 2020 and completed through database access by our research group by August 2020.

 3. We note that you have indicated that data from this study are available upon request. PLOS only allows data to be available upon request if there are legal or ethical restrictions on sharing data publicly. 

The de-identified data used contains potentially identifying patient information, so that the ethical committee of the University Hospital Frankfurt has prohibited the disclosure of the data.

As a contact address for requests to receive the data, the data protection officer of the University Hospital Frankfurt can be contacted: Datenschutz@kgu.de.

'EHA received a research grant of the German Research Foundation (AD 592/1-1)

KZ received financial support from a multitude of companies, a detailed list is attached to the manuscript.

The further authors declare that there are no conflicts of interest. '

The manuscript has been expanded to include the relevant paragraph that the reported "competing interests" does not change compliance with PLOS ONE guidelines for release of data and materials. 

Competing Interests:

EHA received a research grant of the German Research Foundation (AD 592/1-1). This does not alter our adherence to PLOS ONE policies on sharing data and materials.

KZ received financial support from a multitude of companies, a detailed list is attached to the manuscript. This does not alter our adherence to PLOS ONE policies on sharing data and materials.

In order to provide an adequate statement, the cover letter was extended by the sentence:

The authors confirm that the disclosed conflicts of interest of EHA and KZ does not alter our adherence to PLOS ONE policies on sharing data and materials.

5. Please include captions for your Supporting Information files at the end of your manuscript, and update any in-text citations to match accordingly. Please see our Supporting Information guidelines for more information: http://journals.plos.org/plosone/s/supporting-information

The Supporting material was adapted according to the Guidelines and a caption was added at the end of the manuscript.

6. Review Comments to the Author

The authors sincerely thank the reviewers and the editorial office of PLOSOne for their time, courtesy and expert review of our manuscript. We carefully considered their concerns and have altered the manuscript accordingly. We truly believe that attending to these expert critiques/comments has significantly improved the quality of our manuscript. Below please find detailed/point-by-point responses to the reviewers’ questions below:

Reviewer #1: The goal of the authors was to describe sedation practices in patients with COVID ARDS in their medical center. They propose that their patients received unusually high doses of sedatives. Given that this is a descriptive study, I propose that some descriptions need to be extended/more granular, for a practicing clinician to derive value from this study.

Concerns

1. The authors did not compare the COVID-19+ ARDS cohort to cohort with nonCOVID- ARDS to support their claim that COVID+ patients require higher doses of sedatives. Such comparisons are feasible and were already published by other authors (e.g., Wongtangman Epub ahead of print. PMID: 33861551.) The authors need to reference this published study in the Introduction and discuss how their observations are unique from this published study.

We thank the reviewer for providing the reference to the publication of Wongtangman et al.; as the findings had not yet been published at the time of submission, we were not able to include them in our work until now. 

In the revised manuscript, we therefore now refer to this publication and contextualize our work on it. 

“Recently, Wongtangman et al. published a first retrospective comparison between ARDS patients with and without causative COVID-19 pneumonia. They demonstrated a significantly increased need for sedation and analgesics on the basis of a sedative burden index.[Wongtangman] In addition, Kapp et al. were able to demonstrate a significant association between sedation depth and mortality.[Kapp]”

In general, the findings of Wongtangman et al. are in agreement with the data we collected demonstrating a higher utilization of analgesics and hypnotics in mechanically ventilated patients.

However, Wongtangman et al. applied different approaches to analyse their data, which restricts the comparability with our results. Wongtangman et al describe the first retrospective comparison of the administration of sedatives between ARDS patients with and without causative COVID-19 pneumonia. For this purpose, a Sedation Burden Index (SBI) was applied to assign the cumulative burden of sedation. This index seems to be derived from the Drug Burden Index, which was first described in 2018.[7] However, to our knowledge, this Sedation Burden Index has not yet been validated or recommended for use in critically ill patients, and to date, no data are available to support this approach. Moreover, the authors refer to the number of prescriptions of the different sedatives, which certainly demonstrates a higher use in COVID-19 patients but does not allow a comprehensible conclusion on the applied daily dosages.

Thus, our study is the first to provide insight on the requirement for combined use of sedative agents as well as feasible drug combinations with precise dosage of sedative agents.

2. Authors state they they recorded “sedation levels” but these are not presented in the figures or tables. A Figure which would display daily sedation levels with daily doses of sedatives throughout the duration of mechanical ventilation would strengthen the paper. I assume that sedation levels decrease (i.e. RASS moves towards positive values) as survivors approach extubation/tracheostomy but RASS may/may not decrease in nonsurvivors. 

We thank the reviewer for the comment and are pleased to add another graph to our manuscript for better illustration the measured parameters. (Fig 4)

In this figure, daily Richmond Agitation and Sedation Scale scores over the course of observation are shown, as well as deaths within the study population or dropout according to the exclusion criteria.

How do daily drug doses differ in survivors/nonsurvivors as they progress through the ICU stay? Given extremely high mortality in this cohort (48%) such comparison could be feasible. An interesting finding would be relatively low exposure to sedatives, but deep coma, in nonsurvivors – suggesting other factors (e.g. metabolic, inflammatory) in driving the coma.

We thank the reviewer for the valuable note. Regarding the fatalities occurred, we now provide more clarity with the additional graph (Figure 4).

This figure displays 20 of the 26 deaths among our collective out of 56 patients. There were six deaths after the observation period and are thus not shown. 

No difference was observed between survivors/non-survivors during the study period or in the follow-up with respect to the required sedation dosages. 

Also, the interesting question raised by the reviewer regarding the occurrence of lower exposure to sedatives in deep coma in deceased patients was not observed.

The high mortality of 46.6% in our cohort is in line with the internationally observed mortality among intubated ventilated patients. To avoid the impression of an excess mortality in our collective, we have now included a meta-analysis of Lim et al. in the revised manuscript. 

“Our results are consistent with the repeatedly raised suspicion of aggravated sedation with previously published results from other study groups.[Kapp, Wongtangman], which has been raised repeatedly. In our study, we found a high mortality (46.6%) comparable to international data among mechanically ventilated severely affected C-ARDS patients.[Lim] However, we could not demonstrate an association to sedation depth, the number of sedatives applied or the required dosages.”

However, a bias cannot be eliminated due to the supraregional ARDS center accreditation with corresponding allocation of severely affected patients and secondary transfers, for example, for ECMO evaluation. 

3. Authors state that their ARDS patients are primarily transferred patients that were initially managed in other institutions. Were the drug exposures in outside institutions analyzed? At what day of mechanical ventilation were these patients transferred? As patients are “stabilized” in the receiving institution, changes to sedatives are oftentimes made. Additionally, oversedation in outside institution will lead to tolerance and the receiving institution “inherits” opioid-tolerant and hypnotic-tolerant patients. I suppose the authors were able to make many positive adjustments to sedation towards weaning sedation in the first 48 hours after they received the patients. These data would be novel and worth to report.

We thank the reviewer for the objection and would like to clarify the previously misleading statement. 

Of the 56 patients included, 22 patients were admitted from other hospitals.

Secondary admitted patients were mainly, transferred soon after intubation and mechanical ventilation, in 17 cases we admitted the patients within the first 72 hours after initiation of sedation. Transfer was mainly done for the evaluation of more advanced care (e.g. ECMO-treatment) or because of regional capacity overload. Continuous sedation documentation from the transferring hospitals could not be obtained as this was a mono centric study of the University hospital of Frankfurt.

We did not observe oversedation or sedation tolerance in the included patients and consider this unlikely due to the predominantly short external therapy and rapid transfer to our center. In contrast, we were able to stabilize both patients admitted directly to our center and those transferred to our center without any detectable differences in doses and number of sedative and analgesic medications. 

However, to provide more detailed information about the proportion of patients transferred to our center, we have added specific data on the number of patients transferred and the time of transfer:

“The notably rare application of NMBAs might be explained by the fact that the majority of patients to 22 patients were secondarily transferred to our COVID-19 center were secondary, meaning the initial treatment phase took place beforehand. The majority of NMBA applications were conducted to resolve therapy-resistant ventilator asynchronies in spontaneous breathing mode.”

4. Authors report that 96% patients were ventilated in prone position. Authors also state that they targeted deeper levels of sedation (RASS-4) in prone position. This means that even some patients with mild and probably all with moderate ARDS were proned in this center. This may seem aggressive proning approach to some other centers. Did patients with mild and moderate ARDS progress to severe, and that’s why they were proned? This needs to be discussed. Otherwise, the whole study could be viewed as a cohort of aggressive proning and associated/justifiable deep sedation. 

There needs to be more granularity in reporting the sedation doses – prone vs. supine position, daily doses, survivors vs nonsurvivors, etc.

We thank the reviewer for revealing this weakness of the study presentation. All patients in this collective met the criteria for moderate to severe ARDS (oxygenation index <200) during the observation period, resulting in prone positioning. As the reviewer correctly assumes, this is because of a further deterioration of the patients during their ICU-stay. We would like to emphasize, that not all patients received positioning therapy during their ICU-stay and that most patients (84,3%) showed an oxygenation index below 150 before the initiation of prone positioning.

To provide appropriate clarity, we have added the corresponding passages to the manuscript.

“All patients worsened their oxygenation index below <200 during treatment, representing moderate or severe ARDS, leading to prone positioning assuming positive effects. “

"In our study, we found a high mortality (46.6%) comparable to international data among mechanically ventilated severely affected C-ARDS patients.[5] We were not able to demonstrate any association with depth of sedation, the number of sedatives administered or the dosage levels. 

5. Authors use variable terminology “ feasible sedation level”, “adequate sedation level” , “appropriate sedation”, “sufficient analgesia” . The meaning of these is unclear. I propose to use “prescribed sedation level” and “ actual sedation level” . For instance, physician prescribes the level of RASS 0, but on exam, finds that patients has level of RASS-4. 

Analgesia did not seem to be evaluated in this study (e.g. with CPOT scale), therefore it is unclear why authors comment on sufficient analgesia. In general, COVID-19 is not a pain-producing condition unless it is associated with thromboses (MI, mesenteric ischemia, limb ischemia,…).

We would like to thank the reviewer for his objection regarding the various possibly misleading terms and have made adjustments to standardize the reporting in accordance with the reviewer's request.

Moreover, we thank the reviewer to point out that we did not record analgesia in our work, e.g. by using the Critical-Care Pain Observation Tool (CPOT). This was due to the predominantly required deep sedation, e.g., by long periods of prone positioning, so that the CPOT or Behavioral Pain Scale (BPS) could not be consistently recorded due to the lack of interpretability of facial expressions.

Therefore, it was not possible to perform a complete assessment using a validated scale, but analgesia was performed based on bedside assessment and vegetative parameters.

Analgesia was required especially to provide sufficient tube tolerance and for the repeated repositioning and turning maneuvers. In order to make the manuscript more coherent, a corresponding paragraph has been added.

“Due to the predominantly long positioning periods, analgesia monitoring could not be performed continuously using a validated score. Sufentanil was administered primarily to facilitate endotracheal tube tolerance and for positioning maneuvers. We were unable to detect a significant increase in the required sedatives for prone positioning.”

6. Who were the patients that required highest levels of ketamine and sufentanil or the combination of 3 sedatives in this cohort? Were all these ECMO patients? Did all these patients die? If patients with highest sedation doses actually survived, this could teach us that very high sedation requirements may not necessarily be followed by bad outcomes. This would be informative.

In our mixed effects regression analysis, we were unable to demonstrate a statistical association between the need for increased sedation and survival or death. 

Of the six patients undergoing ECMO treatment, four died and the remaining patients were successfully discharged. During ECMO therapy, esketamine therapy was required in four of six cases (corresponding to 42.17% of the total duration of therapy). There was no association between increased esketamine or sufentanil dosages and ECMO treatment. Interestingly, Wongtangman et al. were also able to demonstrate an increased need for esketamine.

7. Authors state that they try to avoid propofol. “Due to the risk of a propofol infusion syndrome during longterm ventilation, propofol was only administered for a maximum of a few days in our patients”. Then they state in Methods “propofol or clonidine and in case of sedation difficulties a combination with midazolam is used”. This manuscript needs to be more consistent, if reader is supposed to understand the sedation practices in author institution.

We thank the reviewer for his objection. For better consistency and clarity, we have modified the manuscript accordingly. 

“Primary sedatives were propofol or clonidine and in case of sedation difficulties a combination with midazolam was used. In the case of primary use of propofol, conversion to clonidine was initiated in the case of a prolonged therapy to avoid propofol infusion syndrome.”

We tried to avoid the use of propofol as much as possible according to the recommendations in the long-term use because of the risk of propofol infusion syndrome.[2] However, at the beginning of treatment, the use of propofol could not be avoided. In the case of the predominant prolonged course of therapy, the therapy was switched from Propofol to Clonidine, unless a therapy with both substances was necessary.

Reviewer #2: Dr. Frankfurt, et al have submitted a retrospective chart review of COVID-associated ARDS patients and the sedation management strategies used for these patients. The manuscript highlights how these patients may require higher than average doses of IV sedatives compared to other ICU patients. The structure of the manuscript is appropriate. The logic is clear and relatively well-characterized. The figures and data analyses are appropriately displayed.

COMMENTS:

1. Please review your manuscript more thoroughly for grammar, punctuation, and syntax errors. There were numerous examples of this throughout the text, especially in the figure descriptions.

We thank the reviewer for the notice and have again carefully revised the manuscript again for grammatical, punctuation, and syntax errors.

2. In the discussion, the analysis of your results is stated well; however I believe some further discussion about how your results compare to others is warranted. 

We are happy to comply with this request. In particular, the new literature source kindly provided by Reviewer 1 allows us to discuss the comparability of our data.

“Few data exist to date to compare in-hospital sedation management strategies in the setting of COVID-19 patient care. However, our in-hospital standards appear to be consistent with previously published regimens for severe COVID-19 patients in terms of the classes of agents used. Previously published reports also describe the predominant use of propofol, benzodiazepines, central α2-agonists and potent opioids, as well as the use of esketamine to achieve the prescribed depth of sedation.”

Additionally, any hypothesis or speculation regarding why higher average doses are required should be elaborated upon further. Is it due to the nature of ARDS or are other pathophysiologic factors to be considered as well?

We are very pleased to comply with the reviewer's request to discuss the hypotheses of aggravated sedation among COVID-19 patients in more detail and have adjusted the manuscript accordingly:

“However, it remains unclear what causes the impaired sedation. Nevertheless, early hyposmia has been described as a characteristic symptom of COVID-19.[6] In the meantime, it was demonstrated that the novel corona virus does not only cause an infection of the lungs, but can also affect the central nervous system in addition to many other organs.[1] Especially in severely affected COVID-19 patients with viremia, an alteration of the CNS is conceivable. Thus, the aggravated sedation could occur as a consequence of an infection of the central nervous system.”

Therefore, like several other centres, we are currently investigating the possibility of detecting cerebral anomalies by electroencephalography in patients with COVID-19 ARDS and sedation. (e.g. NCT04699916, NCT04527198, NCT04405544, NCT04815109)

3. The most common combination of your hospital's IV sedative regimen should be compared to others published in the literature if possible. Otherwise, the generalizability of your results is in question as higher doses may have been required due to non-ideal practices. For instance, other hospitals use dexmedetomidine much more readily than clonidine infusions due to its purported benefits with alpha-2 receptor selectivity and delirium prevention.

Many thanks for this valuable objection regarding the comparability of the hospital standard .

We would like to address the reviewer's objection by describing our sedation regimen in context of the published data:

To date, few data exist to date to compare in-hospital sedation management standards in the setting of COVID-19 patient care. However, our in-hospital standards appear to be consistent with previously published regimens for severe COVID-19 patients in terms of the classes of agents used. Previously published reports also describe the predominant use of propofol, benzodiazepines, central α2-agonists and potent opioids, as well as the use of esketamine to achieve the prescribed depth of sedation. [4; 9]

The substance clonidine is preferred in our clinic, as no direct comparison has yet been able to show a relevant therapeutic advantage of clonidine over dexmedetomidine. Various studies comparing dexmedetomidine with other sedative groups showed a therapeutic advantage, e.g., in comparison with propofol[3], but the SPICE III study published in 2019 could not show a relevant outcome benefit of dexmedetomidine as a sedative compared to standard care. [8]

4. An analysis of other factors that may have caused higher than average mean doses should be considered as well. Did patients have protracted delirium? Did other complications occur that led to more days of sedation?

We would like to meet this valuable objection of the reviewer by going into more detail about the observed complications in the manuscript. 

It should be noted that the observation period of this study was limited to the intensive care period.

“We did not observe any protracted delirium in the patients we observed during the study period. Also, no disproportionate rate of nosocomial infections or cardiovascular complications leading to prolonged sedation were observed.”

Thank you for your submission and your hard work!

[1] Boldrini M, Canoll PD, Klein RS. How COVID-19 Affects the Brain. JAMA Psychiatry 2021.

[2] Hemphill S, McMenamin L, Bellamy MC, Hopkins PM. Propofol infusion syndrome: a structured literature review and analysis of published case reports. British journal of anaesthesia 2019;122(4):448-459.

[3] Hughes CG, Mailloux PT, Devlin JW, Swan JT, Sanders RD, Anzueto A, Jackson JC, Hoskins AS, Pun BT, Orun OM. Dexmedetomidine or propofol for sedation in mechanically ventilated adults with sepsis. New England Journal of Medicine 2021;384(15):1424-1436.

[4] Kapp CM, Zaeh S, Niedermeyer S, Punjabi NM, Siddharthan T, Damarla M. The use of analgesia and sedation in mechanically ventilated patients with COVID-19 ARDS. Anesth Analg 2020.

[5] Lim ZJ, Subramaniam A, Reddy MP, Blecher G, Kadam U, Afroz A, Billah B, Ashwin S, Kubicki M, Bilotta F. Case fatality rates for patients with COVID-19 requiring invasive mechanical ventilation. A meta-analysis. American journal of respiratory and critical care medicine 2021;203(1):54.

[6] Nouchi A, Chastang J, Miyara M, Lejeune J, Soares A, Ibanez G, Saadoun D, Morélot-Panzini C, Similowski T, Amoura Z. Prevalence of hyposmia and hypogeusia in 390 COVID-19 hospitalized patients and outpatients: a cross-sectional study. European Journal of Clinical Microbiology & Infectious Diseases 2021;40(4):691-697.

[7] O'Connell J, Burke É, Mulryan N, O'Dwyer C, Donegan C, McCallion P, McCarron M, Henman MC, O'Dwyer M. Drug burden index to define the burden of medicines in older adults with intellectual disabilities: An observational cross‐sectional study. British journal of clinical pharmacology 2018;84(3):553-567.

[8] Shehabi Y, Howe BD, Bellomo R, Arabi YM, Bailey M, Bass FE, Bin Kadiman S, McArthur CJ, Murray L, Reade MC. Early sedation with dexmedetomidine in critically ill patients. New England Journal of Medicine 2019;380(26):2506-2517.

[9] Wongtangman K, Santer P, Wachtendorf LJ, Azimaraghi O, Baedorf Kassis E, Teja B, Murugappan KR, Siddiqui S, Eikermann M, Group ftSOMT. Association of Sedation, Coma, and In-Hospital Mortality in Mechanically Ventilated Patients With Coronavirus Disease 2019–Related Acute Respiratory Distress Syndrome: A Retrospective Cohort Study. Critical Care Medicine 9000;Latest Articles.

---

## [Editor Report · Decision Letter 1]

14 Jun 2021

High sedation needs of critically ill COVID-19 ARDS patients- a monocentric observational study.

PONE-D-21-08369R1

Dear Dr. Flinspach,

We’re pleased to inform you that your manuscript has been judged scientifically suitable for publication and will be formally accepted for publication once it meets all outstanding technical requirements.

Kind regards,

Chiara Lazzeri

Academic Editor

PLOS ONE
---

## [Editor Report · Acceptance letter]

16 Jul 2021

PONE-D-21-08369R1 

High sedation needs of critically ill COVID-19 ARDS patients- a monocentric observational study. 

Dear Dr. Flinspach:

I'm pleased to inform you that your manuscript has been deemed suitable for publication in PLOS ONE. Congratulations! Your manuscript is now with our production department. 

Kind regards, 

on behalf of

Dr. Chiara Lazzeri 

Academic Editor

PLOS ONE